# Study on Morphological Changes and Interference in the Development of *Aedes aegypti* Caused by Some Essential Oil Constituents

**DOI:** 10.3390/tropicalmed8090440

**Published:** 2023-09-07

**Authors:** Michele Teixeira Serdeiro, Thiago Dutra Dias, Natanael Teles Ramos de Lima, José Maria Barbosa-Filho, Renato de Souza Belato, Jacenir Reis dos Santos-Mallet, Marise Maleck

**Affiliations:** 1Laboratório Interdisciplinar de Vigilância Entomológica em Diptera e Hemiptera, Instituto Oswaldo Cruz, Fundação Oswaldo Cruz, Rio de Janeiro 21040-900, Brazil; 2Laboratório de Insetos Vetores, Campus Vassouras and Campus Maricá, Universidade de Vassouras, Vassouras 27700-000, Brazil; 3Mestrado Profissional em Ciências Ambientais, Universidade de Vassouras, Rio de Janeiro 27700-000, Brazil; 4Laboratório de Tecnologia Farmacêutica, Universidade Federal da Paraíba, João Pessoa 58000-900, Brazil; 5Laboratório de Vigilância e Biodiversidade em Saúde, Universidade de Iguaçu, Nova Iguaçu 26260-045, Brazil; 6Laboratório de Entomologia Médica e Forense, Instituto Oswaldo Cruz, Fundação Oswaldo Cruz, Rio de Janeiro 21040-900, Brazil; 7Colégio Pedro II, Campo de São Cristóvão, Rio de Janeiro 20921-440, Brazil

**Keywords:** *Aedes aegypti*, essential oils, larvicidal activity, transmission electron microscopy

## Abstract

Dengue, Chikungunya and Zika are arboviruses, transmitted by the mosquito *Aedes aegypti*, that cause high mortality and serious health consequences in human populations. Efforts to control *Ae. aegypti* are important for preventing outbreaks of these diseases. Essential oil constituents are known to exhibit many activities, such as their use as larvicides. Given their potential, the present study aimed to characterize the larvicidal effect of dihydrojasmone, p-cymene, carvacrol, thymol, farnesol and nerolidol on the larvae of *Ae. aegypti* and their interference over the morphology of the mosquitos. The essential oil constituents were dissolved in dimethylsulfoxide at concentrations of 1–100 μg/mL and were applied in the breeding environment of third-stage larvae. The larvae from bioassays were fixed, dehydrated and embedded. Ultrathin sections were contrasted using 5% uranyl acetate and 1% lead citrate for observation through transmission electron microscopy. The oil with the highest larvicidal efficiency was found to be nerolidol, followed by farnesol, p-cymene, carvacrol, thymol and dihydrojasmone, with an LC_50_ of 11, 21, 23, 40, 45 and 66 µg/mL, respectively. The treated *Ae. aegypti* larvae caused alteration to the tegument or internal portions of larvae. The present study demonstrated which of these oils—dihydrojasmone, farnesol, thymol, p-cymene, carvacrol and nerolidol—have effective larvicidal activity.

## 1. Introduction

*Aedes aegypti* (Linnaeus 1762) (Diptera: Culicidae) is considered an important vector regarding public health, as it transmits several arboviruses that affect humans. These arboviruses include the Dengue virus, and *Ae. aegypti* is an important vector for urban Yellow fever [1,2]. This mosquito is also a vector for the Chikungunya virus (CHIKV)—the most common symptoms are fever and characteristically severe joint pain—and the Zika virus (ZIKV)—which has been associated with cases of microcephaly among newborns from women infected with this virus, along with other neurological disorders such as the Guillain–Barré syndrome [3,4]. The prevalence of these diseases has been increasing across the world where *Ae. aegypti* is endemic, covering tropical and subtropical areas [5]. About half of the world’s population is now at risk of Dengue with an estimated 100–400 million infections occurring each year in more than 100 countries [2]. 

Vector control programs play an important role in the prevention of mosquito-borne diseases [6]. Therefore, efforts to control the population of the mosquito *Ae. aegypti* are essential for preventing outbreaks of the Dengue, Zika and Chikungunya viruses [6,7]. Current interventions include the elimination of mosquito larvae by applying larvicides to larval breeding sites. In general, the main larvicidal agents are based on organophosphate chemicals and bacterial agents [8]. These agents have undesirable effects, such as toxicity to the environment and to non-target organisms, including humans. [9]. Faced with these challenges, there has been a quest for safer and more sustainable alternatives for arboviral vector control.

One effective intervention is the use of natural plant products that can act as insecticides but are environmentally safe [10,11]. Plant products have been evaluated for their toxic properties against insects, especially in the form of essential oils [12]. Essential oils, also known as volatile oils or ethereal oils, are complex mixtures of volatile compounds, lipophilic, generally odoriferous and liquid, originating from the secondary metabolism of plants [13,14]. They present acute contact and fumigant toxicity to insects, repellent and antifeedant activities, as well as present development and growth inhibitory activity [15]. Thus, essential oils have gained attention as potential bioactive agents against *Ae. aegypti*, either in their crude form or by means of purified substances [16]. Essential oils are mainly composed of terpenoids, particularly monoterpenes and sesquiterpenes [13], and they are known to exhibit many activities; for example, they act as nematicides [17], leishmanicides [18] and insecticides [19]. In addition, they have been widely used for medicinal and cosmetic applications in the pharmaceutical, sanitary, cosmetic, agricultural and food industries [20]. Some oils, such as thymol and carvacrol, are food flavorings that are generally recognized safe (GRAS) and are an indication of low mammalian toxicity for starting materials [21]. In view of these facts, it was of our interest to test essential oils’ constituents on *Ae. aegypti*. Initially, 27 plant oil constituents were screened, and the most successful ones were focused on this study: dihydrojasmone, p-cymene, carvacrol, thymol, farnesol and nerolidol (unpublished study).

The elucidation of the mode of action of natural products in *Ae. aegypti* larvae has a fundamental importance to intensify its effects and for the development of a larvicide. As a tool for this, morphological studies help to understand the toxic effects and possible mechanisms of action [22,23,24]. Thus, the present study aimed to characterize the larvicidal effects of dihydrojasmone, p-cymene, carvacrol, thymol, farnesol and nerolidol on third-stage *Ae. aegypti* larvae and their effects on the morphology of the mosquito using electron transmission microscopy.

## 2. Materials and Methods

### 2.1. Essential Oils’ Constituents

Dihydrojasmone, p-cymene, carvacrol, thymol, farnesol and nerolidol were purchased from Sigma-Aldrich (St. Louis, MO, USA).

### 2.2. Aedes aegypti

*Aedes aegypti* eggs were obtained from the colony of mosquitos that is kept at the Laboratório de Mosquitos Transmissores de Hematozoários, IOC/FIOCRUZ, Rio de Janeiro, RJ, Brazil. Bioassays on larvicidal activity were conducted in the Laboratório de Insetos Vetores/Universidade de Vassouras, Vassouras, RJ, Brazil. Eggs (*n* = 500) were placed in a receptacle (12.5 cm × 8 cm × 2 cm) containing mineral water (1 L) and fish food (0.3 mg/larvae) (Alcon Guppy^®^) for hatching. The receptacle with the eggs was kept in a Bio-Oxygen Demand incubator (BOD) at 27 ± 1 °C, 70 ± 10% relative humidity and 12 h photoperiod. After hatching and development, third-stage larvae (L3) were separated for testing.

### 2.3. Bioassays

The assays to evaluate larvicidal activity were carried out as described by Maleck et al. (2017) [25], adapted from WHO (2005) [26]. The essential oils’ constituents were diluted in dimethylsulfoxide (DMSO) and were applied at final concentrations of 1, 10, 30, 50, 60, 70, 80, 90 and 100 μg/mL in glass containers (4.0 cm × 4.5 cm) with mineral water (20 mL). Twenty third-stage larvae (L3) of *Ae. aegypti* were used in each of three groups: (1) test (essential oils’ constituents diluted in dimethylsulfoxide (DMSO) at concentrations of 1, 10, 30, 50, 60, 70, 80, 90 and 100 μg/mL); (2) control (without essential oils’ constituents and without dilution solvent); (3) DMSO control (testimony) (without essential oils’ constituents and with dilution solvent). The experiments were conducted in triplicate and, thus, there were 60 larvae per group in the three repetitions. The insects were kept in a Bio-Oxygen Demand incubator (BOD) at 27 ± 1 °C, 70 ± 10% relative humidity and 12 h photoperiod. The bioassays were observed regarding their development, viability and mortality.

### 2.4. Statistical Analysis

The results obtained from the biological assays were subjected to Tukey’s test through the GraphPad Prism software, version 6.0 for Windows (GraphPad Software, La Jolla, CA, USA—www.graphpad.com; accessed on 20 August 2017). One-way analysis of variance (ANOVA) was applied using the significance level *p* < 0.01. Significance level according to the Tukey’s test is represented as *p* < 0.0001; *p* < 0.001; *p* < 0.01 and *p* < 0.1 vs. DMSO control. The LC_50_ regression equation and 95% confidence interval value for respective essential oil constituents were determined using Probit analysis. The data were performed in Minitab^®^ 17 Statistical Software (Minitab Inc., State College, PA, USA).

### 2.5. Transmission Electron Microscopy

Transmission electron microscopy was conducted on the larvae from the bioassays of larvicidal activity: (1) test; (2) control; (3) DMSO control. After exposure, the collected larvae (*n* = 3 per group) were fixed in a 2.5% glutaraldehyde solution, in a buffer of 0.1 M sodium cacodylate (pH 7) for one hour at room temperature. After rinsing in the same buffer, the material was post fixed in 1% osmium tetroxide in a 0.1 M sodium cacodylate buffer (pH 7.2), for one hour at room temperature, in the dark. Dehydration was conducted using increasing grades of acetone (50%, 70%, 90% and 100%) followed by infiltration and embedding in Epon 812 epoxy resin. The sections were prepared using an ultramicrotome, contrasted with 5% uranyl acetate and 1% lead citrate for observation. The ultrathin sections were observed using a FEI™, Model Tecnai Spirit iCorr 120 kV transmission electron microscopy at the National Center for Structural Biology and Bioimaging (CENABIO)/Universidade Federal de Rio de Janeiro.

## 3. Results

### 3.1. Larvicidal Activity

Testing of the essential oil constituent dihydrojasmone on *Ae. aegypti* L3–L4 larvae extended the larval period at concentrations of 70 µg/mL (10.1 ± 3.4 days; *p* < 0.001) and 80 µg/mL (11.7 ± 3.6 days; *p* < 0.001) in relation to DMSO control (7.3 ± 0.9 days). This same oil also interfered in the development of L3 larvae to the adult phase at concentrations of 70 µg/mL (11.8 ± 3.7 days; *p* < 0.001) and 80 µg/mL (13.7 ± 3.6 days; *p* < 0.001) while the testimony showed 9.2 ± 1.1 days (Table 1). Dihydrojasmone caused larval mortality of LC_50_ = 66 µg/mL.

The treatment using p-cymene in the breeding environment of *Ae. aegypti* L3 larvae extended the larval development period at concentrations of 1 µg/mL (8.2 ± 1.2 days; *p* < 0.001) and 10 µg/mL (8.5 ± 1.1 days; *p* < 0.0001), in comparison with the DMSO control (7.3 ± 0.9 days). At concentrations of 1 µg/mL (10.2 ± 1.2 days; *p* < 0.01) and 10 µg/mL (10.5 ± 2.3 days; *p* < 0.0001), p-cymene interfered with the L3–adult development period, while this phase of development was shorter in the DMSO control (9.2 ± 1.1 days) (Table 2). Larvicidal potential for p-cymene demonstrated LC_50_ of 23 µg/mL. 

The bioassays on larvae treated with carvacrol showed that this oil extended the larval development period (9.9 ± 3.7 days; *p* < 0.001) and the L3–adult development period (11.9 ± 3.7 days; *p* < 0.0001) at the concentration of 1 µg/mL when compared with the DMSO control (7.8 ± 1.8 days and 10 ± 1.6 days, respectively) (Table 3). The mortality among L3 larvae caused by carvacrol demonstrated an LC_50_ of 40 µg/mL.

Thymol did not interfere with the development period of *Ae. aegypti* (Table 4). The mortality among *Ae. aegypti* L3 larvae treated with thymol was observed in the present study with an LC_50_ of 45 µg/mL of the larvae were affected. 

Farnesol applied in the breeding environment of *Ae. aegypti* larvae decreased the larval development period at a concentration of 30 µg/mL (4.4 ± 0.9 days; *p* < 0.001) and decreased the L3–adult development period at the concentration of 30 µg/mL (6.1 ± 0.7 days; *p* < 0.001) compared with the DMSO control (7.3 ± 0.9 days) (Table 5) The tests showed larvicide activity, and farnesol presented an LC_50_ of 21 µg/mL. Mortality was not observed in the DMSO control and control.

Nerolidol at the concentration of 10 µg/mL extended the larval development period (9.1 ± 2.7 days; *p* < 0.0001) and the L3–adult development period (11.1 ± 2.7 days; *p* < 0.0001) of *Ae. aegypti* larvae, in comparison with the DMSO control (Table 6). The bioassays using nerolidol presented an LC_50_ of 11 µg/mL. Mortality was not observed in the DMSO control and control.

### 3.2. Transmission Electron Microscopy

The analysis on *Ae. aegypti* L3 larvae through transmission electron microscopy in the control and DMSO control (testimony) groups showed that the external cuticular coating was normal, with a laminar appearance and the presence of regular projections on the external surface (Figure 1A). The nuclei of the epithelial cells located just below the cuticle were either oval or presented an irregular appearance, with chromatin distributed uniformly (Figure 1A,B). The mitochondria in this region presented a characteristic double membrane and mitochondrial crests, without any alterations in thickness (Figure 1A,C). The digestive tube presented normal apical, median and basal regions without any morphological alterations, i.e., the epithelial cells were laid out in a single layer of low cylindrical cells, with their apical surface covered by numerous microvilli that were well preserved and elongated (Figure 1D,E). Figure 1E shows the preserved intercellular joints, the homogeneous appearance of the cytoplasm and the presence of abundant mitochondria due to transportation of ions of this type of cell. The muscles presented a typical striated appearance with muscle fibers arranged and ordered parallel to each other (Figure 1F).

Larvae treated with dihydrojasmone presented alterations to the body wall, with cuticle deformations and presence of electron-dense inclusions (Figure 2A–C). Cells presented the severe destruction of cytoplasm and no intact organelles were observed; instead, there were only many myelin figures (Figure 2D).

The ultrastructure of the *Ae. aegypti* L3 larvae treated with p-cymene presented alterations. No cell contours, joints or organelles were observed, and this characterized complete tissue disorganization. The body wall presented cuticle deformations. Moreover, electron-dense inclusions, altered mitochondria and severe vacuolization were also observed (Figure 3A–D).

The treatment with carvacrol resulted in body wall alterations, with cuticle deformations (Figure 4A) and increased proportions of rugose endoplasmic reticulum (Figure 4B). Many myelin figures were found (Figure 4C), and the muscle tissue presented spaces between the fibers (Figure 4D). The cytoplasm was dense, with many ribosomes and granulations and severe vacuolization, and profiles showing tangled endoplasmic reticulum were also seen (Figure 4E,F).

The use of thymol yielded major alterations to the tegument (Figure 5A,B). The cytoplasm presented signs of degeneration, with altered mitochondria (Figure 5C) and severe cytoplasmic vacuolization of varying sizes. The nuclei presented a paler appearance and chromatin fragmentation (Figure 5D). The basal region of the epithelium did not show the typical basal interdigitation appearance, with numerous vacuoles present (Figure 5E). The muscle tissue did not present parallel fiber patterns, and ruptures in the tissue were also observed (Figure 5F).

Treatment with farnesol caused few cuticle alterations. The nuclei of the epithelial cells presented a normal appearance with evident chromatin and nucleolus, while the cytoplasm presented some vacuoles (Figure 6A,B).

The larvae treated with nerolidol presented cuticle deformations in the body wall (Figure 7A). Electron-dense inclusions were also present (Figure 7C), as were altered mitochondria and severe vacuolization. The larvae generally had an altered appearance (Figure 7B), without any visible cell outlines, joints or organelles. The cytoplasm presented a disorganized appearance. Some tracheas were normal in appearance, while others were dilated (Figure 7D). The microvilli were fragmented (Figure 7E). The altered appearance of the cytoplasm was clearly observed, with severe vacuolization and destruction of the cytoplasm (Figure 7F).

## 4. Discussion

The emergence of many arboviral diseases transmitted by *Ae. aegypti* and their capacity to resist synthetic chemical insecticides has increased the interest in exploring new products against this mosquito. Furthermore, extensive usage of synthetic insecticides has caused risks to human health, animals and the environment [27].

Plants are an important and rich source of bioactive chemical compounds that can act effectively towards controlling *Ae. aegypti*, with lower impacts on human health and the environment [10,11].

According to the results from the present study, dihydrojasmone presented high efficacy regarding larvicide activity (LC_50_ = 66 µg/mL) against *Ae. aegypti*, with a duration of 1 to 6 days after the initial application. It is important to show that among the essential oils’ constituents of this study, dihydrojasmone showed the greatest delay in larval development. It should be noted that until now, there had not been any reports in the literature regarding the activity of dihydrojasmone against *Ae. aegypti* larvae. 

In relation to the monoterpene p-cymene, with an LC_50_ of 23 μg/mL. These data demonstrate its larvicidal potential in comparison with p-cymene from *Clausena excavata*, which presented LC_50_ = 43.3 μg/mL against fourth-stage larvae (L4) of *Ae. aegypti* and LC_50_ = 34.9 μg/mL against fourth-stage larvae (L4) of *Aedes albopictus* [28]. 

According to Govindarajan et al. (2016) [29], carvacrol was seen to have larvicidal activity against different species of mosquitos, including *Anopheles stephensi* (LC_50_ = 21.15 μg/mL), *Anopheles subpictus* (LC_50_ = 24.06 μg/mL), *Culex tritaeniorhynchus* (LC_50_ = 27.95 μg/mL) and *Culex quinquefasciatus* (LC_50_ = 26.08 μg/mL). Tang et al. (2011) [30] found similar results for carvacrol against *Aphis craccivora* and *Leucania separata* at median lethal concentrations (LC_50_) of 16.8 and 12.7 μg/mL, respectively. In the present study, carvacrol presented a higher LC_50_ than those mentioned previously (LC_50_ = 40 μg/mL). However, in leaf immersion assays, the action found against *Pochazia shantungensis* nymphs reached LC_50_ = 56.74 μg/mL [31]. 

The larvicide activity of thymol (LC_50_ = 45 μg/mL) found in the present study corroborates what was seen in the study by Waliwitiya et al. (2009) [32] regarding *Ae. aegypti* larvae. It has also been reported that thymol has toxic activity against *Spodoptera litura* [33], *Musca domestica* [34], *Drosophila melanogaster*, *P. shantungensis* [29] and the mosquito *C. quinquefasciatus* [35]. Testing of this monoterpene against the larvae of *An. subpictus*, *Aedes albopictus* and *C. tritaeniorhynchus* mosquitoes gave rise to LC_50_ of 22.06, 24.83 and 28.19 μg/mL, respectively [36]. 

The bioassays with farnesol reported LC_50_ = 21 μg/mL for larvicide activity, while Simas et al. (2004) [37] reported the same activity over *Ae. aegypti* with LC_50_ = 13 μg/mL. In the study by Park et al. (2020) [38], farnesol showed mosquito larvicidal activities and caused retardation of ovarian growth of female *Ae. albopictus* by modulating the formation of the JH receptor complex.

The sesquiterpene nerolidol showed high efficiency, with LC_50_ = 11 μg/mL. This result agrees with what was found in the study by Simas (2004) [37], who reported LC_50_ = 17 μg/mL for *Ae. aegypti*. Larvicidal activity caused by nerolidol was also reported by Chantraine et al. (1998) [39] and Ali et al. (2013) [40] but with LC_50_ = 9 μg/mL and 13.4 μg/mL, respectively. These values were lower than those observed in the present study and in the study by Simas et al. (2004) [37]. Nerolidol isolated from the seeds of *Magnolia denudata* also showed larvicidal activity against the larvae of *Culex pipiens pallens*, *Ae. aegypti*, *Ae. albopictus* and *Anopheles sinensis* [41].

According to the classification by Cheng et al. (2003) [42], the essential oils’ constituents evaluated in the present study, namely farnesol (LC_50_ = 21 µg/mL), p-cymene (LC_50_ = 23 µg/mL), nerolidol (LC_50_ = 11 µg/mL), thymol (LC_50_ = 45 µg/mL) and carvacrol (LC_50_ = 40 µg/mL) are highly active substances for controlling *Ae. aegypti*, since they presented an LC_50_ < 50 μg/mL, while dihydrojasmone (LC_50_ = 66 µg/mL) is merely active since it presented an LC_50_ < 100 μg/mL.

Essential oils frequently show a broad spectrum of bioactivity in relation to the development of insects of importance for public health and agriculture. Their lipophilic nature facilitates their interference in the basic metabolic, biochemical, physiological and behavioral functions of insects [43]. Essential oils can interfere with the feeding behavior of arthropods, act as a growth regulator or act as a neurotoxin, among the toxicity mechanisms; they can also act as protein denaturants and enzyme inhibitors, in addition to promoting the disintegration of the plasma membrane [44,45]. These diverse mechanisms of action contribute to preventing the emergence of resistant vectors. Among the various biological parameters of the effects of essential oils and their constituents on insects, the present study showed the interference of the p-cymene, carvacrol, thymol, farnesol and nerolidol, using electron microscopy, over the morphology of the third-stage larvae (L3).

The larvae treated with the different essential oils’ constituents presented alterations to the external wall of the tegument, thus demonstrating that these substances act directly on the cuticle.

In addition to this aspect, the ultrastructure of the treated larvae also showed alterations to the midgut, with signs of cell destruction and vacuolization of epithelial cells. These signs were indicative of cell disorganization, with spaces between cells and an accumulation of granules in some places in the cytoplasm. There were also pale nuclei, which are characteristic of nuclear degeneration. The microvilli presented an altered appearance, which could hinder food absorption and thus directly affect the nutrition of the larvae. Swollen or destroyed mitochondria hinder ion transportation, which is an important function of epithelial cells. Narciso et al. (2014) [23] evaluated the morphological effect of the neolignan burchellin, isolated from leaves of *Ocotea cymbarum* (Lauraceae), and presented cell disorganization and destruction in the midgut region, spaces between cells and vacuolization of epithelial cells. Maleck et al. (2014) [24] also reported the cytotoxic effect of the amide piperlonguminine, isolated from *Piper tuberculatum* and *Piper acutifolium*, on the epithelial cells of the digestive system of *Ae. aegypti*. That study showed that vacuolization of the cytoplasm and mitochondrial edema occurred.

The severe vacuolization of epithelial cells and the presence of myelin figures indicated cell distress, suggesting the toxicity of essential oils on mosquito larvae. 

The microscopic study conducted by Seye et al. (2021) [46] revealed that the oil of *Cymbopogon citratus* (Lemongrass) acts by causing cell destruction in the larval tissue of *Ae. aegypti* mosquitoes. The study observed significant damage in various cellular structures, including the intestinal microvilli, mitochondria, genetic material, and the body fat mass. According to Cantrell et al. (2010) [47] larvicidal substances can be absorbed through the cuticle of insects, through the respiratory tract, or even through ingestion. Inside the larvae, these substances can reach their site of action or cause systemic effects by means of diffusion in different tissues [48].

## 5. Conclusions

The results of our study demonstrated that the oils’ constituents dihydrojasmone, farnesol, thymol, p-cymene, carvacrol and nerolidol have effective larvicidal activity against *Ae. aegypti*, and among these, nerolidol presented the lowest LC_50_. The importance of the description of dihydrojasmone as an important agent against *Ae. aegypti* larvae needs to be highlighted. The toxicity of dihydrojasmone, farnesol, thymol, p-cymene, carvacrol and nerolidol on *Ae aegypti* larvae was demonstrated with the morphological study, being evidenced in the tegument or internal portions of the larvae and in what indicates their influence in the physiology and morphology of *Ae. aegypti*.

## Figures and Tables

**Figure 1 tropicalmed-08-00440-f001:**
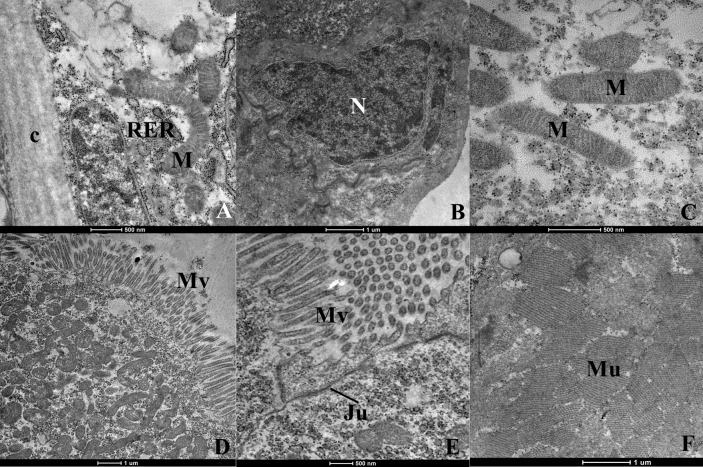
Ultrastructure of *Aedes aegypti* larvae (L3)-group control and DMSO control (testimony). (**A**)—cuticle (c), nucleus (N), endoplasmic reticulum rough (ERR) and mitochondria (M) of normal aspect; (**B**)—nucleus (N) normal; (**C**)—mitochondria (M) normal; (**D**)—apical region of the digestive tube where we visualize microvilli (Mv); (**E**)–region of contact between two cells of the digestive tract demonstrating the integrity of the intracellular junction (Ju); (**F**)–muscles with normal appearance.

**Figure 2 tropicalmed-08-00440-f002:**
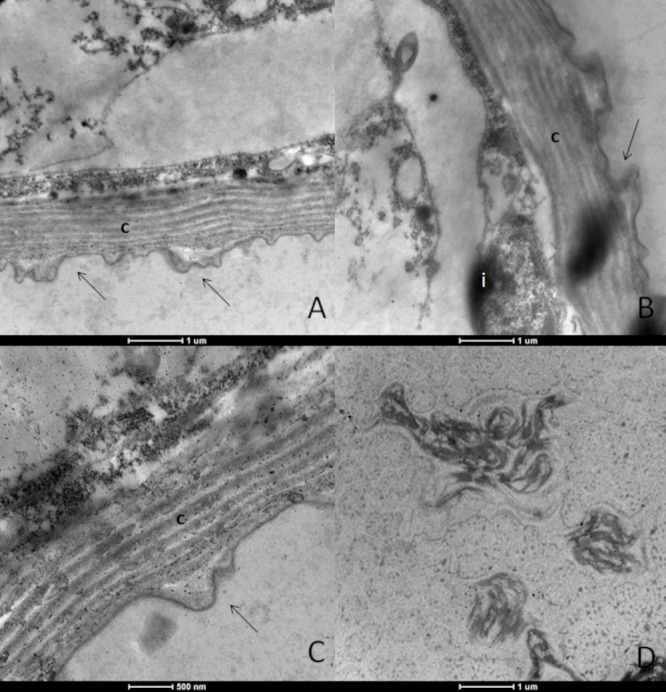
Ultrastructure of *Aedes aegypti* larvae (L3) treated with the substance dihydrojasmone. (**A**–**C**)—body wall showing the cuticle (c) with deformation (arrows); (**B**)—electrodensive inclusions present and intense cytoplasmic destruction; (**D**)—presence of myelin figures.

**Figure 3 tropicalmed-08-00440-f003:**
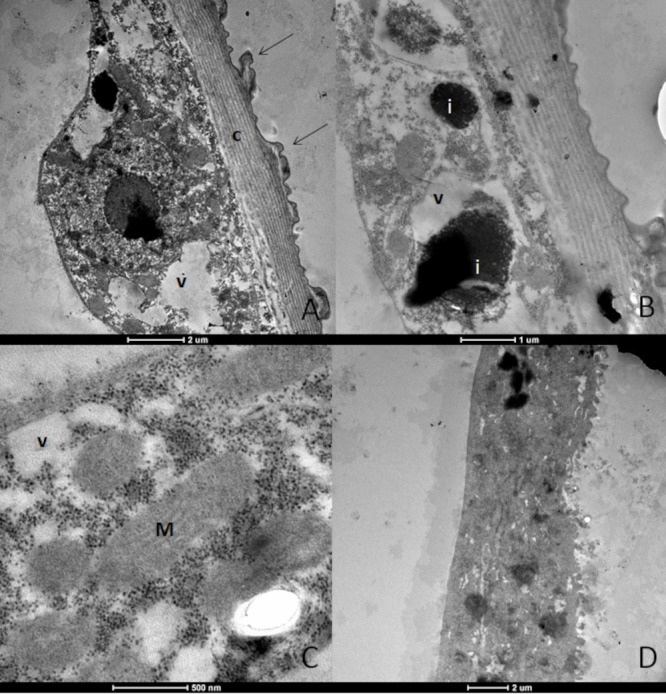
Ultrastructure of *Aedes aegypti* larvae (L3) treated with the substance p-cymene. (**A**)—body wall showing the cuticle (c) with deformation (arrows); (**B**)—electrodensive inclusions present (i); (**C**)—mitochondria (M) with altered appearance and intense vacuolization (v); (**D**)—overview of the larvae demonstrating altered appearance, where the cellular contours, joints and organelles are not observed.

**Figure 4 tropicalmed-08-00440-f004:**
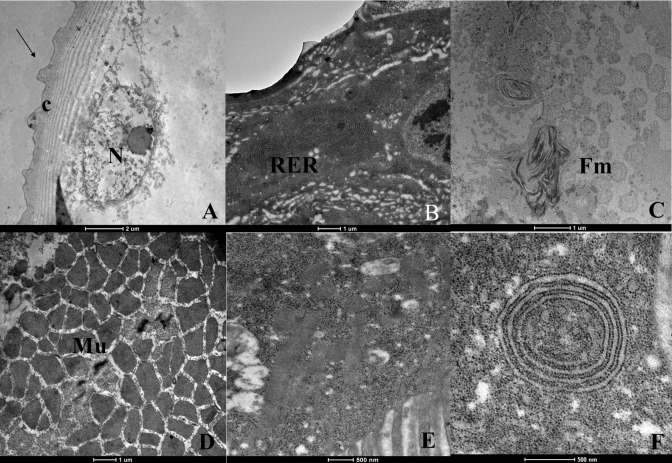
Ultrastructure of *Aedes aegypti* larvae (L3) treated with the substance carvacrol. (**A**)—body wall showing the cuticle (c) with deformation (arrows); (**B**)—endoplasmic reticulum rough (RER) increased; (**C**)—myelin figures (Fm); (**D**)—musculature with spaces between fibers (Mu); (**E**)—dense cytoplasm exhibiting many ribosomes and granulations; (**F**)—large number of ribosomes and endoplasmic reticulum enveloped.

**Figure 5 tropicalmed-08-00440-f005:**
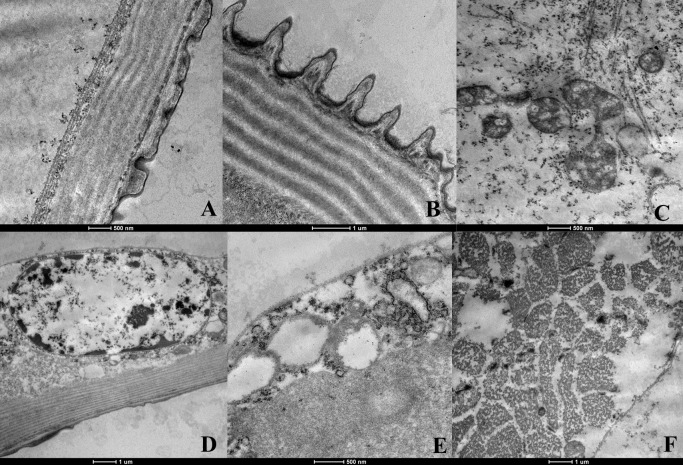
Ultrastructure of *Aedes aegypti* larvae (L3) treated with the substance thymol. (**A**,**B**)—body wall showing the cuticle (c) with deformation; (**C**)—cytoplasm showing signs of degeneration and mitochondria (M) with altered appearance; (**D**)—nucleus (N) with altered appearance; (**E**)—vacuoles (v) in the basal region; (**F**)—muscle cells (Mu) altered.

**Figure 6 tropicalmed-08-00440-f006:**
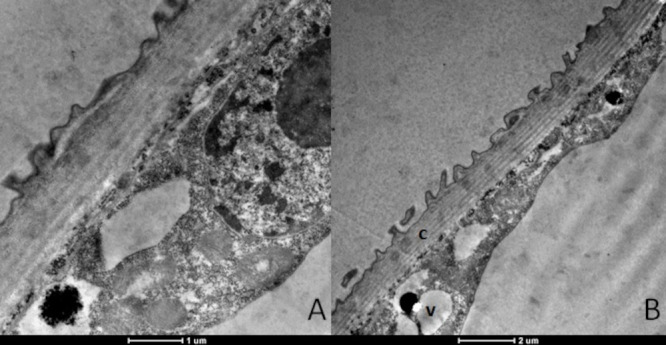
Ultrastructure of *Aedes aegypti* larvae (L3) treated with the substance farnesol. (**A**)—outer portion of tegument showing few alterations in cuticle (c); nucleus (N) altered appearance. (**B**)—cytoplasm showing some vacuoles (v).

**Figure 7 tropicalmed-08-00440-f007:**
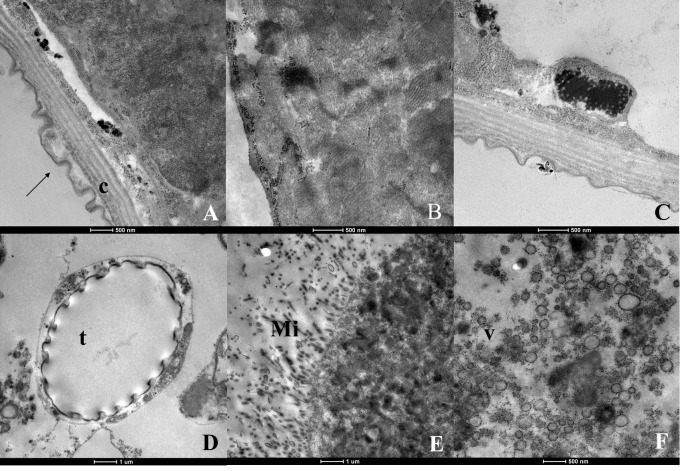
Ultrastructure of *Aedes aegypti* larvae (L3) treated with the substance nerolidol. (**A**)—body wall showing the cuticle (c) with deformation (arrows); (**B**)—view of the larva demonstrating altered appearance, where the cellular contours, joints and organelles are not observed; (**C**)—electrodensing inclusions present; (**D**)—trachea (t) with altered appearance; (**E**)—microvilli (Mi) with fragmented appearance; (**F**)—cytoplasm altered appearance showing intense vacuolization (v) and cytoplasmic destruction.

**Table 1 tropicalmed-08-00440-t001:** Duration of development among *Aedes aegypti* larvae (L3) treated in the middle of farming with dihydrojasmone at concentrations of 1–100 μg/mL.

Treatment	Larvae (Days)		Pupae (Days)		L3–Adult (Days)	
	X ± SD	VI	X ± SD	VI	X ± SD	VI
Control	5.9 ± 1.3 a	4–9	1.8 ± 0.4 a	1–2	7.8 ± 1.5 a	5–11
Testimony	7.3 ± 0.9 b	6–9	2 ± 0 ab	2–2	9.2 ± 1.1 b	6–11
1	6.8 ± 1.5 ab	5–12	2 ± 0 ab	2–2	8.2 ± 1.5 ab	7–14
10	7.9 ± 2.5 b	4–15	2.3 ± 0.3 c ****	2–3	10.2 ± 2.5 ab	6–17
30	6.6 ± 2.2 ab	3–14	2.1 ± 0.7 b	1–3	8.7 ± 2.4 ab	5–16
50	5.9 ± 1.5 a *	2–10	2 ± 0 ab	2–2	7.8 ± 1.4 a *	4–12
60	8.5 ± 2.2 b	4–13	2 ± 0 ab	2–2	10.5 ± 2.2 b	6–15
70	10.1 ± 3.4 c ****	2–17	2 ± 0 ab	2–2	11.8 ± 3.7 c ****	4–19
80	11.7 ± 3.6 c ****	4–18	2 ± 0 ab	2–2	13.7 ± 3.6 c ****	6–20
90	7.7 ± 2.6 b	4–10	2 ± 0 ab	2–2	9.5 ± 3 ab	6–12
100	0	0	0	0	0	0

Experiments with 20 larvae (L3) of *Ae. aegypti*, for each test group and control, in triplicate (*n* = 60). Mean and standard deviation (X ± SD). Range of variation (VI). Values followed by the same letter (a = a, b = b and c = c) did not present any significant difference. Significance level according to the Tukey test is represented as **** *p* < 0.0001 and * *p* < 0.1 vs. DMSO control (testimony).

**Table 2 tropicalmed-08-00440-t002:** Duration of development among *Aedes aegypti* larvae (L3) treated in the middle of farming with p-cymene at concentrations of 1–80 μg/mL.

Treatment	Larvae (Days)		Pupae (Days)		L3–Adult (Days)	
	X ± SD	VI	X ± SD	X ± SD	VI	X ± SD
Control	5.9 ± 1.3 a	4–9	1.8 ± 0.4 a	1–2	7.8 ± 1.5 a	5–11
Testimony	7.3 ± 0.9 b	6–9	2 ± 0 ab	2–2	9.2 ± 1.1 b	8–11
1	8.2 ± 1.2 c ***	4–10	1.9 ± 0.3 ab	1–2	10.2 ± 1.2 c **	5–12
10	8.5 ± 1.1 c ****	6–10	2 ± 0 ab	2–2	10.5 ± 2.3 c ****	8–12
30	5.6 ± 1.3 a ****	4–8	1.9 ± 0.3 ab	1–2	7.5 ± 1.4 a ****	5–10
50	7.3 ± 1.8 b	6–10	2 ± 0 ab	2–2	9 ± 1.6 ab	8–12
60	5 ± 0 ab	5–5	2 ± 0 ab	2–2	7 ± 0 ab	7–7
70	6 ± 0 ab	6–6	1 ± 0 c ****	1–1	7 ± 0 ab	7–7
80	0	0	0	0	0	0

Experiments with 20 larvae (L3) of *Ae. aegypti*, for each test group and control, in triplicate (*n* = 60). Mean and standard deviation (X ± SD). Range of variation (VI). Values followed by the same letter (a = a, b = b and c = c) did not present any significant difference. Significance level according to the Tukey test is represented as **** *p* < 0.0001; *** *p* < 0.001 and ** *p* < 0.01 vs. DMSO control (testimony).

**Table 3 tropicalmed-08-00440-t003:** Duration of development among *Aedes aegypti* larvae (L3) treated in the middle of farming with carvacrol at concentrations of 1–60 μg/mL.

Treatment	Larvae (Days)		Pupae (Days)		L3–Adult (Days)	
	X ± SD	VI	X ± SD	X ± SD	VI	IV
Control	7.8 ± 1.8 a	4–12	2.1 ± 0.4 a	1–3	9.8 ± 1.7 a	6–13
Testimony	7.8 ± 1.8 ab	4–11	2.2 ± 0.5 ab	1–3	10 ± 1.6 ab	7–13
1	9.9 ± 3.7 c ***	3–18	2 ± 0 ab	2–2	11.9 ± 3.7 c ****	5–20
10	7.9 ± 3.2 ab	4–16	2 ± 0.3 ab	2–2	10.3 ± 3.2 ab	6–18
30	7.9 ± 2.5 ab	4–14	2.2 ± 0.5 ab	2–4	9.9 ± 2.4 ab	6–18
50	5.1 ± 1.1 d	4–7	3.2 ± 1 c ****	2–4	8.3 ± 1.2 ab	7–11
60	0	0	0	0	0	0

Experiments with 20 larvae (L3) of *Ae. aegypti*, for each test group and control, in triplicate (*n* = 60). Mean and standard deviation (X ± SD). Range of variation (VI). Values followed by the same letter (a = a, b = b, c = c and d = d) did not present any significant difference. Significance level according to the Tukey test is represented as **** *p* < 0.0001 and *** *p* < 0.001 vs. DMSO control (testimony).

**Table 4 tropicalmed-08-00440-t004:** Duration of development among *Aedes aegypti* larvae (L3) treated in the middle of farming with thymol at concentrations of 1–80 μg/mL.

Treatment	Larvae (Days)		Pupae (Days)		L3–Adult (Days)	
	X ± SD	VI	X ± SD	VI	X ± SD	VI
Control	6.6 ± 3 a	3–14	2.3 ± 0.5 a	2–3	10 ± 2.9 a	6–18
Testimony	7.7 ± 3.2 ab	3–16	1.9 ± 0.3 b	1–2	10.2 ± 3.4 ab	6–18
1	8.1 ± 2 b	5–15	2 ± 0 b	2–2	10.3 ± 2.1 ab	7–16
10	7.5 ± 1.1 ab	6–9	2.2 ± 0.4 a	2–3	9.7 ± 1.2 ab	8–12
30	7.6 ± 1.5 ab	3–10	2 ± 0 b	2–2	9.8 ± 1.5 ab	5–12
50	6.6 ± 0.8 ab	5–7	2.2 ± 0.4 a	2–3	8.7 ± 0.9 ab	7–10
60	5.4 ± 1.1 ab	4–7	2 ± 0 b	2–2	7.7 ± 1.2 ab	6–9
70	5 ± 1 ab	4–6	2 ± 0 b	2–2	6.7 ± 1.5 ab	5–8
80	0	0	0	0	0	0

Experiments with 20 larvae (L3) of *Ae. aegypti*, for each test group and control, in triplicate (*n* = 60). Mean and standard deviation (X ± SD). Range of variation (VI). Values followed by the same letter (a = a and b = b) did not present any significant difference.

**Table 5 tropicalmed-08-00440-t005:** Duration of development among *Aedes aegypti* larvae (L3) treated in the middle of farming with farnesol at concentrations of 1–90 μg/mL.

Treatment	Larvae (Days)		Pupae (Days)		L3–Adult (Days)	
	X ± SD	VI	X ± SD	VI	X ± SD	VI
Control	5.9 ± 1.3 a	4–9	1.9 ± 0.3 a	1–2	7.8 ± 1.5 a	5–11
Testimony	7.3 ± 0.9 b	6–9	2 ± 0 ab	2–2	9.2 ± 1.1 b	6–11
1	7.3 ± 2 b	4–12	1.9 ± 0.3 ab	1–2	9.3 ± 2.1 b	6–14
10	6.6 ± 2 ab	3–11	2 ± 0 ab	2–2	8.6 ± 1.9 ab	5–13
30	4.4 ± 0.9 ac ***	3–6	1.7 ± 0.5 ab	1–2	6.1 ± 0.7 ac ***	5–7
50	4 ± 1 ac *	3–5	2.2 ± 0.5 ab	2–3	6.3 ± 1.5 ab	5–8
60	5.7 ± 1.2 ab	5–7	2 ± 0 ab	2–2	7.7 ± 1.2 ab	7–9
70	4.5 ± 0.5 ab	4–5	1.5 ± 0.7 ab	1–2	6.5 ± 0.5 ab	6–7
80	5 ± 0 ab	5–5	0	0	0	0
90	6 ± 0 ab	6–6	2 ± 0	2–2	8 ± 0 ab	8–8
100	0	0	0	0	0	0

Experiments with 20 larvae (L3) of *Ae. aegypti*, for each test group and control, in triplicate (*n* = 60). Mean and standard deviation (X ± SD). Range of variation (VI). Values followed by the same letter (a = a, b = b and c = c) did not present any significant difference. Significance level according to the Tukey test is represented as *** *p* < 0.001 and * *p* < 0.1 vs. DMSO control (testimony).

**Table 6 tropicalmed-08-00440-t006:** Duration of development among *Aedes aegypti* larvae (L3) treated in the middle of farming with nerolidol at concentrations of 1–60 μg/mL.

Treatment	Larvae (Days)		Pupae (Days)		L3–adult (Days)	
	X ± SD	VI	X ± SD	VI	X ± SD	VI
Control	5.9 ± 1.3 a	4–9	1.8 ± 0.4 a	1–2	7.8 ± 1.5 a	5–11
Testimony	7.3 ± 0.9 b	6–9	2 ± 0 ab	2–2	9.2 ± 1 b	8–11
1	6.9 ± 1.4 b	4–12	2 ± 0 ab	2–2	9 ± 1.4 b	6–14
10	9.1 ± 2.7 c ****	5–13	2 ± 0 ab	2–2	11.1 ± 2.7 c ****	7–15
30	3 ± 0 a *	3–3	2 ± 0 ab	2–2	5 ± 0 a *	5–5
50	5 ± 0 ab	5–5	2 ± 0 ab	2–2	7 ± 0 ab	7–7
60	0	0	0	0	0	0

Experiments with 20 larvae (L3) of *Ae. aegypti*, for each test group and control, in triplicate (*n* = 60). Mean and standard deviation (X ± SD). Range of variation (VI). Values followed by the same letter (a = a, b = b and c = c) did not present any significant difference. Significance level according to the Tukey test is represented as **** *p* < 0.0001 and * *p* < 0.1 vs. DMSO control (testimony).

## Data Availability

Data and script are available under request.

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
