# Peer review of "Study on Morphological Changes and Interference in the Development of Aedes aegypti Caused by Some Essential Oil Constituents"

_tropicalmed, 2023, doi:10.3390/tropicalmed8090440_

Round 1

Reviewer 1 Report

Moderate editing of English language required.

Author Response

ANSWERS

Study on morphological changes and interference in the development of Aedes aegypti caused by some essential oil constituents

Referee 1:

Paper “Study on morphological changes and interference in the development of Aedes aegypti

caused by some essential oil constituents” by Michele Teixeira Serdeiro et al. aimed to

characterize the larvicidal effect of dihydrojasmone, p-cymene, carvacrol, thymol, farnesol and

nerolidol on the larvae of Ae. aegypti and their interference over the morphology of the

mosquitos. The manuscript was well written and contains promising data, arranged in a

scientific manner. However, the manuscript needs major revision before considering publication

in Tropical medicine and infectious disease.

1- The introduction should be containing more background about the main point and why not

to prepar nanomaterial from this oils to become more effective.

Authors: Done.

2- At the result what does it mean development period and weres mortality percentage and

weres table or histograme for this data of mortality.

Authors: A table was included in the manuscript.

3- the scientific name should be etalic like Ae. Aegypti in line 265 to become Ae aegypti

please revesion to all text.

Authors: Done.

4- The manuscript needs deep discussion.

Authors: Done.

5- Please rephrase the conclusion to contain the promising results.

Authors: Done.

6- Why to select this species only of pathogenic bacteria.

Authors: we have not tested pathogenic bacteria.

7- The image of result need to more resolution to be come more clear if possible.

Authors: Done.

8- I also suggest revising the English language in order to increase the readability of the

manuscript.

Authors: Done.

Reviewer 2 Report

The manuscript by Serdeiro et al is well-written and the results were described in excellent form, albeit some modifications are still required.

Authors tested a number of essential oil constituents to check whether they have impact (larvicidal activity and larva period length) on mosquito larvae and observe the organelle changes/alterations.

line 74-75; Initially 27 plant oils constituents were screened --> any preliminary studies? add reference or add (unpublished study) if the authors did the prelim studies but no publication yet

line 135-167 please consider using a table to display results. Authors may keep these paragraphs or highlight only the significant differences.

Figure 1 to 7. Please insert A-F symbol in a box or other markings for better illustration

line 265, 269, 272 italicize Ae aegypti

line 335 our study?

line 337-338 "The importance of description of dihydrojasmone as an important agent against Ae. aegypti larvae needs to be highlighted"

Please describe in which element did this essential oil showed superiority as it was not mentioned in the discussion

Major Comments; 

Abstract; Why only the larvicidal activity was mestioned in the abstract (with nerolidol being the highest) but in the conclusion, the delayed larval development (dihydrojasmone being the most superior) was emphasized instead. Please mention and describe both of them in both sections

Result and Discussion; I am wondering about the [macro] morphology of the mosquito larva before the ultrathin sections for organelle observation using electron transmission microscopy. Did the authors observe any changes in outside appearances, for example the difference in larval sizes?

If there is, please add in the discussion as it may help the readers.

none

Author Response

ANSWERS

Study on morphological changes and interference in the development of Aedes aegypti caused by some essential oil constituents

Referee 2:

The manuscript by Serdeiro et al is well-written and the results were described in excellent form, albeit some modifications are still required.

Authors tested a number of essential oil constituents to check whether they have impact (larvicidal activity and larva period length) on mosquito larvae and observe the organelle changes/alterations.

  1. line 74-75; Initially 27 plant oils constituents were screened --> any preliminary studies? add reference or add (unpublished study) if the authors did the prelim studies but no publication yet.

Authors: Preliminary data for the 27 essential oil constituents has not been published. See manuscript.

  1. line 135-167 please consider using a table to display results. Authors may keep these paragraphs or highlight only the significant differences.

Authors: A table was included in the manuscript.

  1. Figure 1 to 7. Please insert A-F symbol in a box or other markings for better illustration

Authors: Done. See figure.

  1. line 265, 269, 272 italicize Ae aegypti

Authors: Done.

  1. line 335 our study?

Authors: Yes, our study. See manuscript.

  1. line 337-338 "The importance of description of dihydrojasmone as an important agent against Ae. aegypti larvae needs to be highlighted".

Please describe in which element did this essential oil showed superiority as it was not mentioned in the discussion.

Authors: Dihydrojasmone did not show superior results when compared to other oils. The importance of the description is due to not having found in the literature the description of the larvicidal activity in Aedes aegypti.

Major Comments; 

  1. Abstract; Why only the larvicidal activity was mestioned in the abstract (with nerolidol being the highest) but in the conclusion, the delayed larval development (dihydrojasmone being the most superior) was emphasized instead. Please mention and describe both of them in both sections

Authors: The importance of the description is due to not having found in the literature the description of the larvicidal activity in Aedes aegypti.

  1. Result and Discussion; I am wondering about the [macro] morphology of the mosquito larva before the ultrathin sections for organelle observation using electron transmission microscopy. Did the authors observe any changes in outside appearances, for example the difference in larval sizes?

If there is, please add in the discussion as it may help the readers.

Authors: We did not observe any external changes in the treated larvae.

Round 2

Reviewer 1 Report

The authors revised the manuscript according to reviewers comment and the manuscript can be accepted in the current form